# Associations of habitual physical activity and carotid-femoral pulse wave velocity; a systematic review and meta-analysis of observational studies

Rebecca Lear[1]*, Brad Metcalf[1], Gemma Brailey[1], Michael Nunns[2], Bert Bond[3], Melvyn Hillsdon[1], Richard Pulsford[1]

1 Department of Public Health and Sport Sciences, University of Exeter, Exeter, Devon, United Kingdom, 2 University of Exeter Medical School, Exeter, Devon, United Kingdom, 3 Children's Health and Exercise Research Centre, University of Exeter, Exeter, Devon, United Kingdom

* r.lear2@exeter.ac.uk

**Data Availability Statement:** All relevant data are within the paper and its Supporting Information files.

## Abstract

The aim of this review was to understand the association between habitual physical activity (hPA) and carotid-femoral pulse wave velocity (cfPWV) in an ostensibly healthy adult population. Searches were performed in MEDLINE Web of Science, SPORTDiscus and CINAHL databases published up to 01/01/2022 (PROSPERO, Registration No: CRD42017067159). Observational English-language studies assessing the relationship between cfPWV and hPA (measured via self-report or device-based measures) were considered for inclusion in a narrative synthesis. Studies were excluded if studying specific disease. Studies were further included in pooled analyses where a standardised association statistic for continuous hPA and cfPWV was available. 29 studies were included in narrative synthesis, of which 18 studies provided sufficient data for pooled analyses, totalling 15,573 participants. A weak, significant, negative correlation between hPA and cfPWV was observed; partial r = -0.08 95%CI [-0.15, -0.01]; P = 0.045. Heterogeneity was high ($I^2$ = 94.5% P<0.001). Results did not differ across sub-group analyses, however the high heterogeneity within pooled analyses was largely explained by studies utilizing self-reports of PA exposures, being of poor methodological quality or providing only univariate analyses. Overall this systematic review identified a weak negative beneficial association between hPA and cfPWV suggesting that higher levels of hPA benefit vascular health even amongst an asymptomatic population. However, the variation in PA metrics reported (restricting ability to complete meta-analysis), and the heterogeneity within pooled analyses suggests that findings should be interpreted with a degree of caution. The development of methods to precisely quantify day-to-day movement behaviours should support future high-quality research in this field.

## Introduction

Cardiovascular diseases (CVD) are a major contributor to global mortality. In the UK, 27% of deaths in 2019 were attributable to CVD, of which 43,251 deaths were premature (<75 years

**Funding:** The author(s) received no specific funding for this work.

**Competing interests:** The authors have declared that no competing interests exist.

of age) [1]. An estimated 7.6 million adults in the UK are currently living with CVD representing a significant burden on quality of life [1].

The cardio-protective effects of physical activity (PA, defined as any bodily movement produced by skeletal muscles that results in energy expenditure [2]) have been well established [3]. The relationship between physical activity and CVD follows a curvilinear dose-response relationship. Those with the lowest levels of activity are at the highest risk of CVD [3]. Public health guidelines recommend a minimum of 150 minutes per week of moderate/vigorous intensity PA (MVPA) [4] and meeting this standard is associated with up to 34% (unadjusted) or 23% (adjusted for body weight) reduced risk of CVD mortality and events [5].

The inverse relationship between PA and CVD risk persists after accounting for established risk factors including blood pressure, adiposity, glucose and lipid metabolism, suggesting that additional protective mechanisms exist [6]. Vascular dysfunction is an important preceding mechanism in the development of CVD [7] and has emerged as an additional risk factor, possibly accounting for unexplained PA-related CVD risk reduction.

A number of different techniques have been established to measure vascular health which fall into two distinct, yet related physiological entities; endothelial function, and arterial stiffness. The gold-standard measurement of arterial stiffness, carotid-femoral pulse wave velocity (cfPWV) [8], is highly reproducible, non-invasive and predicts future cardiovascular events and all-cause mortality in an asymptomatic population independent of conventional cardiovascular risk factors [8].

A growing body of research indicates that exercise results in acute [9, 10] and chronic improvements in arterial stiffness [11–15]. Exercise, while an important determinant of health, is a subset of PA characterised by activities that are planned, structured, and repetitive with the objective of improving or maintaining physical fitness [2]. It is undertaken only by a minority of adults [16], and typically represents a small proportion of total daily PA [17]. Evidence for the vascular benefits of structured exercise, although important, offer limited insight into the effects of habitual PA (hPA) which also includes non-exercise leisure time, occupational activity, active travel, and activities of daily living. MVPA accumulated sporadically or within lighter intensity hPA bouts represent a large proportion of habitual MVPA (27% to 98% respectively) [18]. Therefore, interventions promoting increases in hPA rather than only structured exercise may have greater potential to reduce CVD risk across all population groups. However, the independent association between hPA and cfPWV is currently unclear.

Only one published systematic review [19] specifically evaluates associations between hPA and cfPWV with searches conducted up to December 2016. However the interpretation of these results are limited by; 1) the inclusion of studies investigating these associations in disease states such as diabetes and hypertension which are known to affect arterial stiffness [20] and 2) the inclusion of univariate correlations within pooled analyses which do not take into consideration the important effect of confounding variables such as age and sex [21, 22]. Additionally, the inclusion of only studies employing device-based measures of PA ignores a number of important studies in which information on PA was collected using self-report. This restricts the number of studies available for synthesis, but importantly prevents comparison of associations from studies that have employed different methods for assessing the exposure.

Consequently, the purpose of this systematic review is to resolve the above limitations and to provide an updated appraisal of the evidence for the association between hPA and cfPWV in an adult population free from chronic disease.

## Methods

This review was conducted according to best practise [23], and is reported here in accordance with the Preferred Reporting Items for Systematic reviews and Meta-Analyses (PRISMA) [24].

The review protocol was registered on the International Register of Systematic Reviews (PROSPERO, Registration No: CRD42017067159) prior to initiation of literature searches. Initial protocol registration specified inclusion of studies utilising any measure of vascular function as part of a wider PhD thesis, but the outcome measure was restricted to only cfPWV to better allow evidence synthesis.

## Search

Systematic searches were completed up to 01/01/2022 in the following databases: MEDLINE (Ovid) 1946—Present and supplemented with MEDLINE (Ovid) in process and other non-indexed citations, Web Of Science—Core collection, (Including conference proceedings citation index, and emerging sources citation index), SPORTDiscus through EBSCOhost, and CINAHL (Cumulative Index to Nursing & Allied Health Literature) through EBSCOhost. A base search strategy was developed with an information specialist using MEDLINE (S1 File) and syntaxes altered accordingly for each subsequent database search. Supplementary searching was then completed through forward and backward citation searching of included papers to identify any other potentially relevant publications. The search had no date limit but was limited to studies published in the English language.

## Inclusion / Exclusion criteria (PECOS)

**Population.**    Adults (≥18 years) who were free from any chronic cardiovascular, endocrine or metabolic disease. Studies containing both children/adolescents and adults were considered for inclusion if findings from adult populations were reported separately. Studies specifically investigating participants with established chronic conditions were excluded as these disease states would likely alter the vascular response to PA [20], however data from a healthy control group was utilised if reported separately.

**Exposure.**    The exposure of interest was hPA collected using either self-report or device-based measures. Studies focusing on exercise or exercise training, or including no data regarding other non-exercise PA, were excluded.

**Comparators.**    All statistical comparisons assessing the relationship between hPA and cfPWV were included in the narrative synthesis and vote count. Associations of a continuous measure of hPA were included in pooled analyses where appropriate.

**Outcome.**    The outcome of interest was arterial stiffness measured via cfPWV.

**Study design.**    Eligible studies included cross-sectional and/or prospective analyses of associations between free-living hPA and cfPWV. Intervention studies reporting the associations between free-living PA and cfPWV data at baseline were considered for inclusion as cross-sectional analyses.

## Study selection

Records identified from the database searches were examined for eligibility independently by two reviewers (RL and GB) in two stages. A third arbiter was consulted in the case of disagreements between the two reviewers on eligibility of a study at each stage of screening. Title and abstracts were reviewed in stage 1, and potentially relevant studies proceeded to stage 2 where full-texts were obtained and screened against eligibility criteria. The screening criteria were independently piloted by both reviewers on a subset of studies prior to the initial full searches to ensure they were interpreted correctly.

Eligible studies proceeded to data extraction. Forward and backward citation searches were then completed on the included studies to identify any other potentially relevant studies for screening.

## Data extraction

The primary data extracted from each eligible study were the statistical associations between hPA and cfPWV and any covariates included in the analytical models. Additionally, descriptive data was extracted including authors, title, aim, study design, population characteristics, recruitment procedures, participant eligibility criteria, PA measurement method, cfPWV measurement method, and participant preparation prior to cfPWV measurement. Data extraction was cross-checked by reviewer two, and any discrepancies resolved.

## Quality assessment

Quality of the included studies was independently assessed by two reviewers (RL and GB) during data extraction, using the National Institute of Health "Quality Assessment Tool for Observational Cohort and Cross-Sectional Studies" [25] adapted to include assessment of the quality of exposure measurement (hPA). This quality assessment tool consists of 14 questions which assess; study design, selection bias, information bias for exposure (hPA) and outcome (cfPWV), and treatment of confounders. Each study was given a "yes", "no", "not reported" or "cannot determine" for each question and from this rated overall "good" "fair" or "poor" quality. No studies were excluded based on the quality score, instead ratings were used to inform interpretation of findings.

## Data analysis

Studies reporting an overall association between cfPWV and hPA were considered for inclusion in meta-analysis. Due to the variation in hPA metrics reported, only studies providing a reported (or imputed as described below) *standardised* association statistic (partial correlation coefficient; partial r, or standardised beta coefficient; std β) [26] for either Total PA (TPA) (all hPA regardless of intensity) or MVPA (hPA only above a moderate intensity threshold) adjusted for standard covariates including at least age, sex, body mass index (BMI) and blood pressure (BP) were pooled to produce a single summary estimate of the independent association between hPA and cfPWV. We contacted the corresponding authors of 14 studies to request any information required for meta-analysis that was not reported in their papers. If the authors were not contactable or not able to provide this information, where possible, the required data was imputed as follows:

1) In two studies [27, 28] an accurate P-value was obtained from N (sample size) and SE (standard error) utilising the below formula [29] whereby z represents the z-score (z = estimate/ SE)

$$p = \exp(-0 \cdot 717z - 0 \cdot 416z^2)$$

2) In four studies [27, 28, 30, 31] non-standardised associations (b) were converted to partial r, given N and P utilising the below formula [32] whereby t represents the test statistic (t = b / SE).

$$t = \frac{r}{\sqrt{\frac{1-r^2}{N-2}}}$$

3) In eight studies [33–40] univariate correlation values were converted to partial r adjusting for age, sex, BMI, BP (here on referred to as "standard covariates") utilising the following partial r formula [32]:

$$r_{yx_1 \cdot x_2} = \frac{r_{yx_1} - r_{yx_2} r_{x_1 x_2}}{\sqrt{(1 - r_{yx_2}^2)(1 - r_{x_1 x_2}^2)}}$$

whereby $r^2 yx_2$ is the combined association of cfPWV and covariates (estimated as $r^2 = 0.29$), and $r^2 x_1 x_2$ is the combined association of hPA and standard covariates (estimated as $r^2 = 0.06$) from existing literature. On average this conversion of crude r to partial r reduced association estimates by 0.1 which was a similar reduction in association to that reported within included studies which provided both the unadjusted r and partial r.

The standardised association statistics (partial r or std β) were converted to a standard normal metric (Fisher's z scale) for overall effect size calculation in order to account for any differences between exposure and outcome metrics. Fisher's Z and 95% confidence intervals were converted back to correlations for presentation.

Statistical heterogeneity was assessed using I-squared analysis. An $I^2 < 40\%$ was considered "not important" as per the Cochrane recommendations [41]. A 'leave-one-out' sensitivity analysis was completed to explore the influence of individual studies on overall association by removing one at a time from pooled analyses, and additionally for any statistical outliers. Subgroup analyses and meta-regression plots were additionally completed to explore heterogeneity further. A funnel plot and Egger test was completed to identify any risk of publication bias. All analysis was completed in STATA version 17 (StataCorp. 2021. Stata Statistical Software: Release 17. College Station, TX: StataCorp LLC).

All analyses (including those not suitable for inclusion in pooled analyses) were summarised using a vote count approach and discussed in a narrative synthesis. Vote counting quantifies the direction of results based on their positive, negative or non-significant results.

## Results

### Study characteristics

Database searches yielded 8149 studies of which 180 full-texts were screened for eligibility. 151 studies were excluded for the following reasons: Investigating clinical population without healthy control group (N = 21), Measuring exercise only with no measurement of habitual PA (N = 50), alternative measurement of vascular function without cfPWV (N = 51), insufficient data reported (N = 37) and the inadequate categorization of participants restricting analysis (N = 8). Note, a number of studies were excluded for meeting more than one of these exclusion criteria. A PRISMA flow diagram is presented in Fig 1.

Overall, 29 studies met the criteria for inclusion. Three studies were prospective [27, 42, 43] and the remainder cross-sectional (n = 26). Four studies [34, 44–46] included a group or subset of participants with disease, however only data in healthy control group participants were utilised. Sample size varied from 22 to 5184 with an average of n = 862 ± 1379. Overall the combined sample from included studies was 54% male, with two studies including only females [39, 47] and seven including only male participants [33–35, 38, 48–50]. Mean age varied from 21 to 78 years (overall mean 51 ± 7.3 years). Four studies included only participants ≥65 years [43, 48, 49, 51] and three included only participants <45 years [34, 52, 53]. The remaining studies included a range of ages.

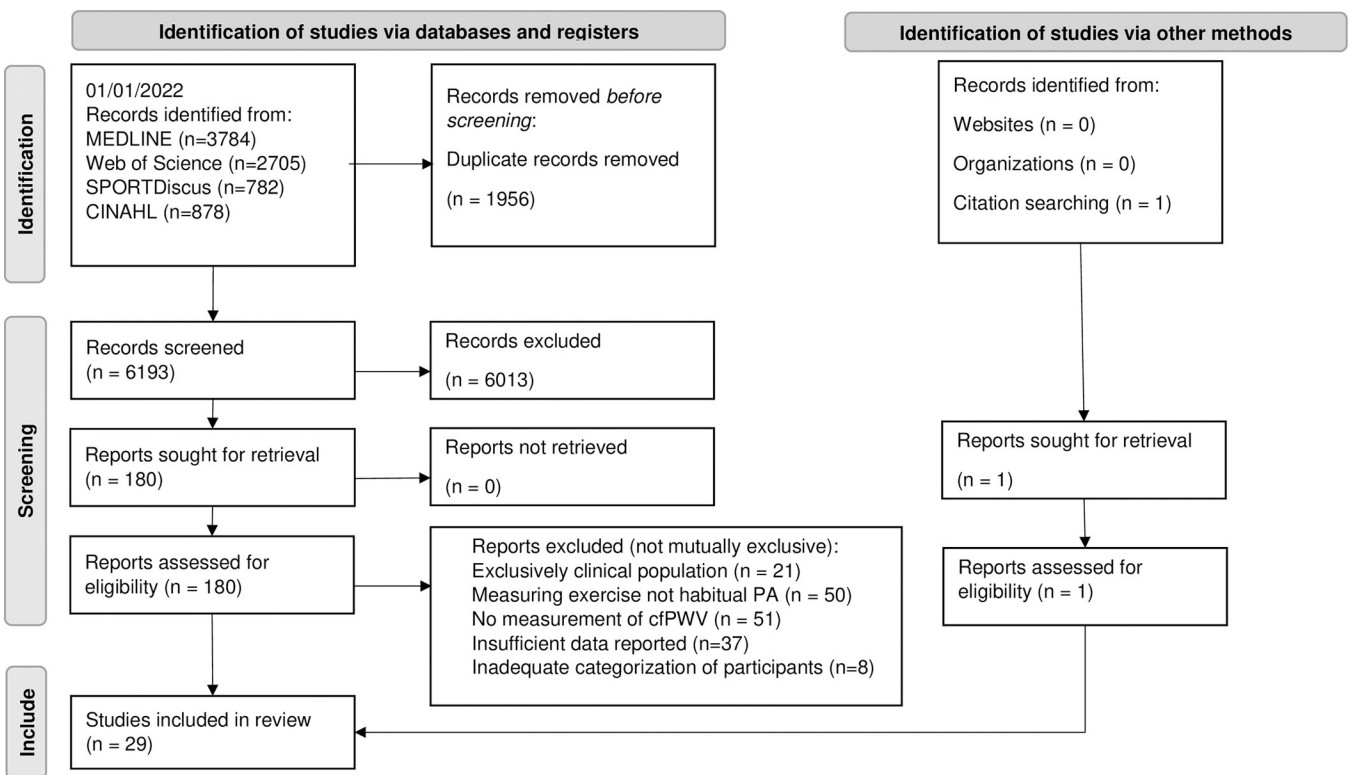

**Fig 1. Preferred reporting items for systematic reviews and meta-analyses (PRISMA) flow diagram of search process.** cfPWV, carotid femoral Pulse Wave Velocity. PA, Physical activity.

Fifteen studies measured hPA exposure using accelerometers [28, 30, 31, 33, 36, 37, 44, 45, 47, 49, 51–55], and 16 measured hPA via self-report questionnaire or interview [27, 30, 34–36, 38–40, 42, 43, 46, 48, 50, 56–58]. Two studies [30, 36] utilised both self-report and accelerometer to measure hPA reporting results separately. The majority of studies (n = 24) measured cfPWV using the tonometry method, and 5 oscillometric method [34, 35, 37, 49, 51]. The most commonly used device (n = 16) was the "SphygmoCor" (ATCor Medical, Australia) [27, 28, 30, 33, 38, 40, 42, 44–46, 48, 52, 53, 55, 57, 58]. 18 studies reported the mean baseline cfPWV in the cohort, ranging from 5.2m/s [53] to 10.71 m/s [47] with an overall mean cfPWV of 8.33 m/s.

Seven studies were given a "good" quality rating [27, 28, 30, 31, 43, 46, 49] whilst thirteen were "fair" [33, 37–40, 44, 45, 48, 51–53, 55, 57] and nine "poor" quality [34–36, 42, 47, 50, 54, 56, 58]. The main sources of bias were questions relating to study design due to the majority of included studies being cross-sectional, poor reporting of participation and follow-up rates, absence of sample size or power calculation, and inadequate blinding. A full data extraction summary can be seen in S1 Table.

## Association of hPA and cfPWV

Of the 29 studies identified for inclusion in this review, a standardised continuous association (via partial r or std β) between hPA and cfPWV, adjusted for at least age, sex, BP and BMI was obtained (or imputed as described in the methods) from 18 studies and thus were included in a pooled analysis. Studies reporting results for both accelerometer and self-report derived PA were averaged for overall pooled analyses and reported separately for subgroup analyses

according to PA measurement method. A summary of study characteristics included in meta-analysis can be found in Table 1. Overall, across a total of 15,573 participants, there was a weak [59], significant, negative correlation between hPA and cfPWV; partial r = -0.08 95%CI [-0.15, -0.01]; P = 0.045 (S1 Fig). However heterogeneity was considerable [41] at $I^2$ = 94.5% P<0.001 necessitating further exploration of possible sources of heterogeneity.

Leave-one-out sensitivity analyses were completed whereby one study is removed from overall meta-analysis at a time. The overall association was not significantly changed by any one study included in pooled analyses, however one study; Kakiyama et al., (1998) [35] identified as having the largest effect on overall association (although not significant) reduced the overall association; partial r = -0.04 95%CI [-0.09, -0.00] (S2 Fig). Removal of this study from main meta-analysis marginally reduced heterogeneity, however this remained significant (P<0.001) and substantial ($I^2$ = 79.0%), S3 Fig.

Subgroup analyses were completed to further explore the heterogeneity of association between studies. In subgroup analyses comparing hPA measurement method (Fig 2), heterogeneity remained substantial in self-reported hPA ($I^2$ = 96.9%) but was ameliorated in studies measuring hPA via accelerometer ($I^2$ = 0.01%). The association with cfPWV in accelerometer measured hPA was weaker but more significant than the association in self-reported hPA (accelerometer; (partial r = -0.06 95%CI [-0.08,-0.04]), self-report; (partial r = -0.18 95%CI [-0.26, 0.11])). However, the difference between the groups was not statistically significant (test of group differences P = 0.87).

Similarly, heterogeneity varied considerably within subsequent subgroup analyses and was considered "not important" in studies of a good quality ($I^2$ = 0.11%, partial r = -0.04 95%CI [-0.06, -0.02]) compared to studies considered fair quality ($I^2$ = 84.5%, partial r = -0.04 95%CI [-0.12, 0.05]) or poor quality ($I^2$ = 92.8%, partial r = -0.31 95%CI [-0.59, 0.05]) Test of group differences P = 0.33, and in studies adjusting for standard + extra covariates ($I^2$ = 0.6%, partial r = -0.04 95%CI [-0.06, -0.03]) compared to studies adjusting only for standard covariates ($I^2$ = 92.5%, partial r = -0.12 95%CI [-0.26, 0.02]) Test of group differences P = 0.29 (Fig 3).

Heterogeneity remained substantial in subgroup analyses comparing the different hPA metrics reported; MVPA ($I^2$ = 50.6% partial r = -0.06 95%CI [-0.10, -0.03]), vs TPA ($I^2$ = 95.3% partial r = -0.09 95%CI [-0.22, 0.05]) Test of group differences P = 0.85, and in subgroup analysis of cfPWV measurement method; tonometry ($I^2$ = 67.0% partial r = -0.02 95%CI [-0.06, 0.01]), vs oscillometric ($I^2$ = 93.6% partial r = -0.30 95%CI [-0.52, -0.04]) however groups were significantly different (P = 0.04) (Fig 3).

Meta-regression revealed that the mean age(4a), sample size(4b), and mean cfPWV(4d) values did not influence the magnitude of the association between hPA and cfPWV (P = 0.213, P = 0.464 and P = 0.356 respectively). However, it did reveal a negligible, yet significant association with sex(4c) indicating that studies with a higher percentage of males tended to report stronger, more negative associations between hPA and cfPWV. Per 1% increase in proportion of males, partial r was reduced by 0.003 (B = -0.003 P = 0.018) (Fig 4).

Egger's test was not significant (P = 0.06) and the funnel plot did not indicate asymmetry indicating there were no small-study effects (S4 Fig).

Analyses were repeated on raw values obtained from the published studies to establish whether there was an effect of the Authors manual conversion of univariate correlation to partial r (as described in the methods). Studies reporting univariate r had a larger (more negative) effect size than studies reporting partial r (univariate r = -0.22 95%CI [-0.39,-0.05], $I^2$ = 88.6%, partial r = -0.05 95%CI [-0.07, -0.03], $I^2$ = 7.4%), S5 Fig. Subgroup analysis for PA measurement method remained similar; (accelerometer; (r = -0.06 95%CI [-0.09,-0.04]), self-report; (r = -0.14 95%CI [-0.31, 0.03]), S6 Fig).

**Table 1. Summary of 18 studies included in pooled analyses.**

| Author | Year | Country | Sample size N (%male) | Age (M ±SD) | PA Measure | Average cfPWV (m/s) | Covariates | Study Quality | Association included in Meta-Analysis |
|---|---|---|---|---|---|---|---|---|---|
| Kakiyama [35] | 1998 | Japan | 139(100%) | 40 ± 14 | S-R | 6.7 | Standard# | Poor | **Total PAEE (kcal/week) partial r = -0.51423, P<0.001** |
| Kakiyama [34] | 1999 | Not Specified | 28(100%) | 21 ± 2 | S-R | 5.5 | Standard# | Poor | **Total PAEE (kcal/week) partial r = -0.63180, P<0.001** |
| Ronnback [38] | 2007 | Finland | 54(100%) | 58 (52–78) | S-R | 7.9 (median) | Standard# | Fair | Total MET-hr/week partial r = 0.021, P = 0.428 |
| Kozakova [36] | 2013 | Not Specified | 45(51%) | 42 ± 9 | Acc | 8.4 | Standard# | Poor | Average counts/min partial r = -0.102, P = 0.166 |
| | | | | | S-R | 8.4 | Standard# | Poor | Total MET-min/week partial r = 0243 P = 0.948 |
| Crichton [42] | 2014 | USA | 505(40%) | 61 ± 12 | S-R | 10.5 | Standard | Poor | Total MET-hr/week standardised β = -0.032 P = 0.413 |
| Gomez-Marcos [30] | 2014 | Spain | 263(41%) | 56 ± 12 | Acc | 7.1 (median) | Standard + Extras | Good | Average counts/min partial r = -0.041, P = 0.51## |
| | | | | | S-R | | Standard + Extras | Good | Total MET-hrs/week partial r = -0.116 P = 0.06## |
| Andersson [31] | 2015 | England | 2376(46%) | 47 ± 9 | Acc | 7.1 | Standard + Extras | Good | **MVPA (min/day) partial r = -0.056, P = 0.006##** |
| Ayabe [51] | 2015 | Japan | 206(47%) | 73 ± 5 | Acc | 11.9 | Standard + Extras | Fair | **MVPA (min/day) partial r = -0.295, P = 0.004** |
| Horta [52] | 2015 | Brazil | 1171(51%) | 30 ± 0 | Acc | Not Reported | Standard + Extras | Fair | **MVPA (min/day) Standardised β = -0.074, P = 0.015** |
| Laursen [28] | 2015 | Denmark | 1404(51%) | 66 (61–71) | Acc | Not Reported | Standard + Extras | Good | **PAEE (kJ/kg/day) partial r = -0.055, P = 0.041##** |
| Mac Ananey [37] | 2015 | Ireland | 79(65%) | 39 ± 9 | Acc | 6.9 | Standard# | Poor | Relative MVPA (min/day) partial r = 0.009, P = 0.292 |
| Parsons [49] | 2016 | Britain | 1118(100%) | 78 ± 5 | Acc | 10.2 | Standard + Extras | Good | MVPA (10min/day) standardised β = -0.042 P>0.05 |
| Ahmadi-Abhari [27] | 2017 | UK | 5184(73%) | 65 ± 6 | S-R | 8.4 | Standard + Extras | Good | **MVPA (hr/week) partial r = -0.030, P = 0.031##** |
| Calvacante [33] | 2019 | Portugal | 98(100%) | 55 ± 7 | Acc | 10.1 | Standard# | Fair | **MVPA (min/day) partial r = -0.089 P = 0.048** |
| Stamatelopoulos [39] | 2020 | Greece | 625(0%) | 58 ± 8 | S-R | 8.9 | Standard# | Poor | Total PA (min/day) partial r = 0.101 P = 0.261 |
| Vandercappellen [55] | 2020 | Netherlands | 1242(43%) | 60 ± 8 | Acc | 9.1 | Standard + Extras | Fair | Total PA (hr/week) Standardised β = -0.04, P = 0.193 |
| Fernberg [53] | 2021 | Sweden | 658(27%) | 22 ± 2 | Acc | 5.3 | Standard | Fair | Total PA (min/day) Standardised β = -0.074, P = 0.059 **MVPA (min/day) Standardised β = -0.079, P = 0.042** |
| Islam [40] | 2022 | USA | 378(40%) | 53 ± 10 | S-R | 7.6 | Standard# | Fair | Total PA (score) partial r = 0.11, P = 0.56 |

PA, physical activity. S-R, self-report. Acc, accelerometer. PAEE, Physical activity energy expenditure. MET, metabolic equivalents. MVPA, moderate to vigorous physical activity. Age; represented as mean age ± standard deviation or Median (inter-quartile range) rounded to 0d.p. Average cfPWV represented as sample mean, or (median).

Covariates; Standard includes age, sex, body mass index and blood pressure. Standard + Extra includes any number of additional covariates included in analysis as reported in original study.

\# indicates reported crude values were manually adjusted by authors of this review to adjust for standard covariates; age, sex, body mass index and blood pressure.

\#\# indicates non-standardised association statistic reported was manually converted to standardised partial r by authors of this review.

Significant associations are indicated in **bold**

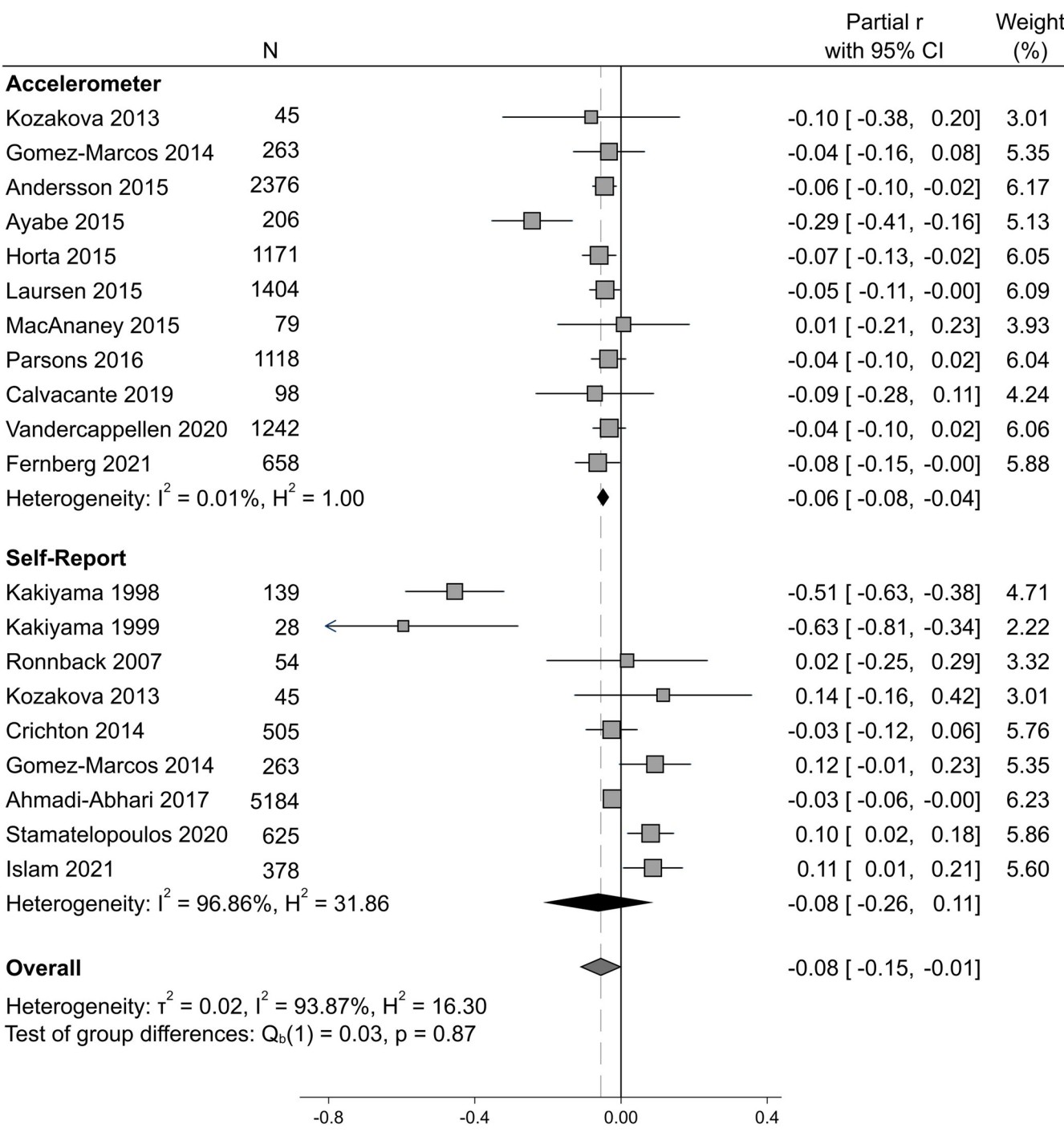

**Fig 2. Forest plot of pooled analysis indicating the association between habitual moderate-vigorous or total physical activity and carotid-femoral pulse wave velocity, grouped by physical activity measurement method.** Random effects model. Grey squares indicate individual study association with 95% confidence intervals (CI). Solid vertical line represents line of no difference. Dashed vertical line and grey diamond represents the overall summary estimate of association. Black diamonds represent the individual subgroup overall effect size.

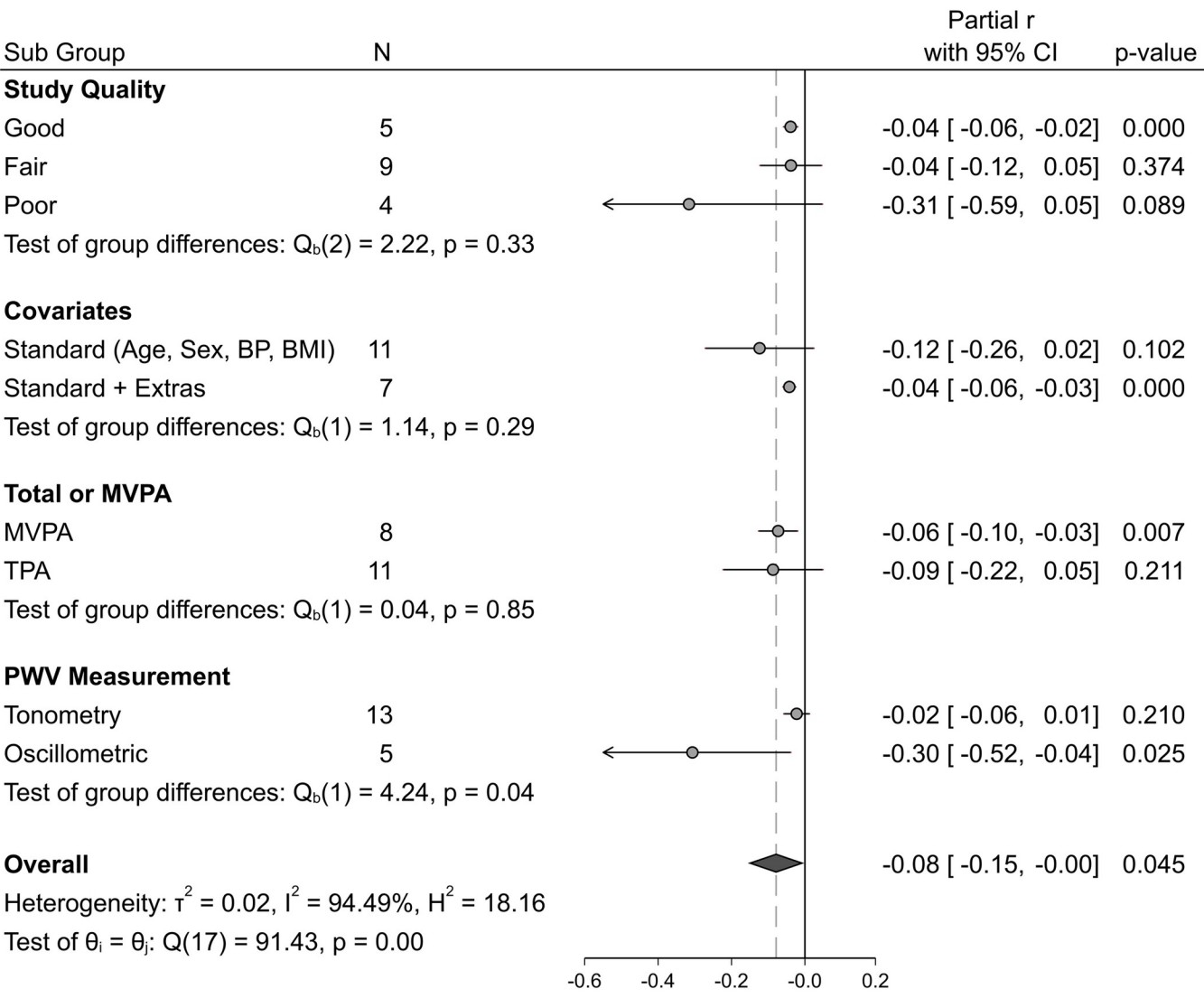

**Fig 3. Sub-group summary meta-analysis indicating the association between habitual physical activity and carotid-femoral pulse wave velocity.** Stratified by different subgroups: Study Quality (good vs fair vs poor), Covariates included in analysis (standard vs standard + extras), physical activity metric included in analysis (total vs moderate-vigorous PA), and the Pulse Wave Velocity measurement method (Tonometry vs Oscillometric). Random effects model. Grey circles indicate summary association estimate for each subgroup with 95% confidence intervals (CI). Solid black vertical line represents line of no difference. Dashed grey vertical line and solid grey diamond represents the overall summary estimate of association. Standard covariates includes age, sex, blood pressure and body mass index. Standard + extras covariates could include any number of additional covariates as reported within individual studies.

## Vote count summary

Of the 90 statistical analyses reported across the 29 included studies, 38 reported a negative (beneficial) relationship between hPA and cfPWV, 52 reported no significant association, and no analyses reported a positive (detrimental) relationship with PA and cfPWV (Table 2). A greater proportion of beneficial associations was observed in studies of good/fair quality compared to poor (49% vs 31% significant associations), and in those reporting adjustments for standard covariates compared to univariate analyses (52% vs 31% significant associations).

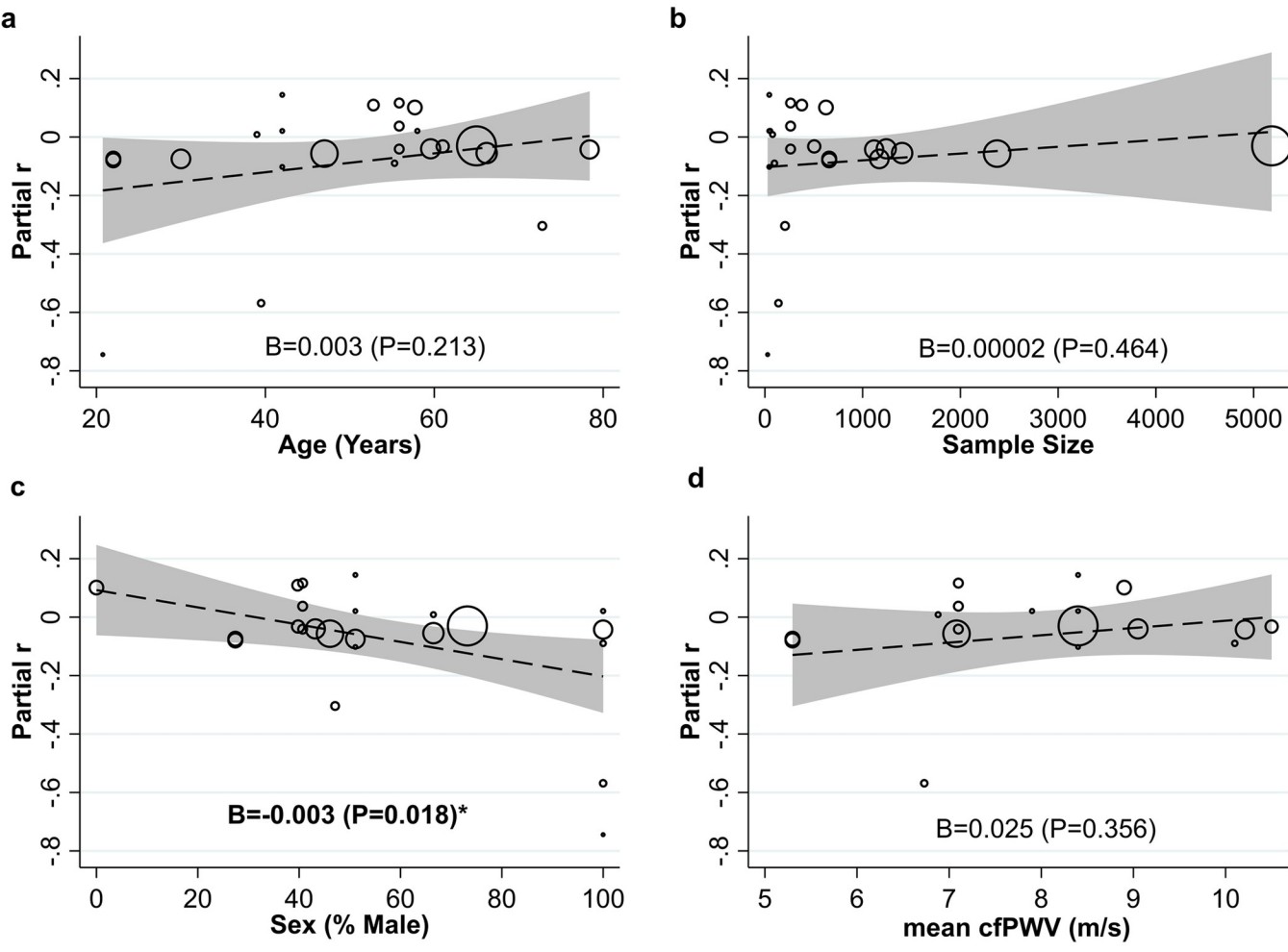

**Fig 4.** Bubble plots and meta-regression for mean participant age (4a) sample size (4b) sex (% male) (4c) and mean cfPWV value (m/s) (4d) against standardised association statistic between hPA and cfPWV. Circles represent individual studies, dashed line represents linear prediction and shaded area 95% CI. *indicates significant association in bold.

## Discussion

The aim of this systematic review was to synthesise and evaluate available evidence for the association between hPA and cfPWV in an ostensibly healthy adult population, providing an update whilst addressing some of the limitations of a previous systematic review [19]. Pooled data indicates a weak, statistically significant, beneficial association between hPA and cfPWV (partial r = -0.08 95%CI [-0.15, -0.01]; P = 0.045). This is the first meta-analysis to demonstrate such a relationship with free-living hPA, which includes occupational activities, active travel and PA associated with daily living as well as structured exercise in an exclusively healthy population.

Germano-Soares et al (2018) reported a considerably greater overall association of -0.16 95%CI [-0.26, -0.06]; P<0.01 between MVPA and cfPWV [19]. The difference in magnitude of effect between the two reviews is unsurprising given that Germano-Soares et al included studies of disease states such as hypertension which are known to accentuate the association between PA and cfPWV [20]. Additionally, Germano-Soares included both adjusted and unadjusted correlation coefficients in the same pooled analyses which potentially inflated the

**Table 2. Vote counting across all included studies.**

|  | ↓ | | ↑ | | ↔ | | Total |
|---|---|---|---|---|---|---|---|
|  | n(%) | | n(%) | | n(%) | | n |
| cfPWV Total | 38 | (42) | 0 | (0) | 52 | (58) | **90** |
| Sub-group vote count | | | | | | | |
| Prospective | 4 | (67) | 0 | (0) | 2 | (33) | **6** |
| Cross-sectional | 34 | (40) | 0 | (0) | 50 | (60) | **84** |
| Accelerometer | 23 | (42) | 0 | (0) | 32 | (58) | **55** |
| Self-report | 15 | (43) | 0 | (0) | 20 | (57) | **35** |
| Adjusted analyses | 25 | (52) | 0 | (0) | 23 | (48) | **48** |
| Univariate analyses | 13 | (31) | 0 | (0) | 29 | (69) | **42** |
| Good / fair quality | 27 | (49) | 0 | (0) | 28 | (51) | **55** |
| Poor quality | 11 | (31) | 0 | (0) | 24 | (69) | **35** |

cfPWV, carotid femoral pulse wave velocity. n, number of vote counts

↓ significant negative (beneficial) association

↑ significant positive (detrimental) association

↔ no association

independent association between hPA and cfPWV. We know from our own sensitivity analyses that studies only reporting univariate analyses reported a markedly higher association than those adjusted for important covariates; univariate r = -0.22 compared to a partial r of -0.05 in the present analyses. This highlights the importance of rigorous methodology in ensuring any potential confounding factors are taken into consideration in order to elucidate the independent association between hPA and cfPWV.

## Clinical significance of findings

The observed pooled association between a lower cfPWV and higher levels of hPA may have important clinical implications even amongst a healthy adult population. A recent meta-analysis including nearly 20,000 participants indicated that per 1 m/s increase in cfPWV there was a 12% increase risk of CVD events and a 9% increased risk of CVD mortality [60]. In the Framingham Heart Study [61] a 1SD higher baseline cfPWV was associated with a 48% increased risk of first major CVD event within the follow-up period (median 7.8 years) amongst 2200 previously CVD-free adults, after adjustment for traditional CVD risk factors [61]. Although the association between hPA and cfPWV indicated in the present review was weak, our results show that per 1SD increase in hPA, cfPWV is reduced by 0.08SD. Based on the Framingham Heart Study data this reduction in cfPWV could be translated to a ~4% decreased risk of first CVD event amongst an adult population free from chronic disease.

## Methodological considerations

In the present review, statistical heterogeneity within overall pooled analyses was high, and varied considerably between sub-group analyses. Of particular note, is that heterogeneity was high amongst studies assessing PA via self-report, studies of a poor methodological quality and those adjusting only for standard covariates. Indeed, 14 of the 29 included studies reported no confounding variables were taken into consideration in statistical analyses. Amongst these, within our vote-count analysis, 42 statistical comparisons were made, of which 31% indicated a significant beneficial association, and 69% indicated no significant association. By contrast within 48 statistical comparisons in which standard covariates were taken into consideration

52% reported a significant beneficial association of hPA and 48% non-significant. This suggests the impact of confounding variables may have influenced the attenuation to null in these analyses.

**Measurement and reporting of physical activity.** Generally, the methods for collecting PA exposure data were well defined and implemented consistently across participants. However, there was considerable variation in how authors measured and reported hPA. In order to compare studies with differing hPA measurement units, all association results were converted to standardised β or partial r which allowed us to compare per standard deviation change in hPA and cfPWV and thus allowing direct comparisons across differing measurement units. Despite this, within pooled analyses heterogeneity was substantially different in studies measuring PA via self-report ($I^2$ = 96.9%) compared to those utilising accelerometers ($I^2$ = 0.01%). Although it is unlikely that the difference in heterogeneity was fully explained by PA measurement alone (as studies utilising accelerometers tended to be of higher methodological quality and included a larger number of covariates within analyses) results do suggest that PA measurement method may have had an influence on the variation in pooled analyses.

This in part may be due to the variation in self-report measures utilised in different studies, which estimate different domains of PA (frequency, intensity, duration and type of PA) requiring different levels of detail from respondents and from which summary PA metrics would be calculated differently. Conversely, although data processing methods may vary, the accelerometers employed in included studies all primarily measured movement acceleration, regardless of manufacturer.

Self-reports are also often subject to substantial error, particularly for lower intensity or short / intermittent activities which may be difficult to recall with precision. This may impact individual as well as group PA quantification and classification [62]. Misclassification of volume and intensity of PA, if non-differential, would lead to the attenuation of any true association with cfPWV towards the null. However it must also be recognised that, misclassification is also possible in studies reporting accelerometer-estimated PA derived utilising acceleration thresholds for MVPA applied uniformly for all participants, which assumes that all activity accumulated above the threshold is of equal value and does not take into account an individual's exercise capacity [63].

Despite the above limitations, self-reports of PA continue to make an important contribution to large scale population level research given their relative cost, feasibility, and low-participant burden [62]. They can also provide information on the type of activity (i.e. occupational, household, transport etc) and relative intensity, both of which are difficult to capture with accelerometers [62]. As such while the limitations associated with self-reports and the differences in measures used may have contributed to substantial between-study variance, the inclusion of these studies in the meta-analysis, and comparison with studies employing device-based measures is an important strength of this review.

**Assessment and reporting of cfPWV.** Subgroup analyses revealed a significant difference (p = 0.04) between studies that utilised the tonometry (measured) method (partial r = -0.02) in comparison to those utilising the oscillometric (estimated) method (partial r = -0.30). This is despite previous research demonstrating a good agreement between the two measures, and thus both are considered valid measures of cfPWV [64]. It must be noted that of the five studies within the oscillometric subgroup, three were of poor quality and reported only univariate associations (necessitating the manual conversion to partial r, as reported in the methods), of which one was the article identified in the leave-one-out analysis as having the largest influence on outcome (Kakiyama et al., 1998) [35].

Finally, the present review identified a negligible, but significant, effect of sex (%males) on pooled analyses suggesting that the association between hPA and cfPWV may be stronger in

males. Previous research has shown that the menopause may reduce any association of PA on arterial stiffness [65]. Bia and Zocalo (2021) identified a cfPWV 'sex x age' interaction of a magnitude r = -0.0087 95%CI [-0.0146, -0.0028] P = 0.0041 in 1289 participants aged 3–84 years [22]. This suggests a 'sex x age' interaction (pre/post-menopausal) within the present review, however further research would be needed to confirm this interaction.

## Future directions

This review has highlighted a number of important observations regarding associations of PA and cfPWV; firstly the substantial heterogeneity in methodological quality of included studies may have affected overall results. This highlights the need for good quality research in order to continue to develop our understanding of the impact of hPA on arterial stiffness and CVD risk.

Secondly, the hPA metric reported within studies varied considerably restricting comparisons. Studies assigning uniform intensity thresholds to all participants is likely to lead to misclassification of PA intensity. The inclusion of only MVPA in some studies is a limitation in itself as it ignores the potential contribution of lighter intensity activities. Light PA is an important component of total habitual activity and constitutes the largest proportion of free-living hPA, particularly in the elderly. Accelerometers are sensitive to very small changes in movement and are able to accurately characterise day-to-day activities that are short in duration and light in intensity. As a result it is important for future research to collect data on total hPA rather than focussing only on physical activities above or within a certain threshold. In a recent consensus statement authors suggest that although a single recommendation on the analytical approaches to accelerometer derived PA cannot be made, analyses should investigate more detailed PA intensities and patterns than typically studied [66].

Finally, most studies included in this review only focus on aggregate data in the form of average volumes of PA (either total or separately according to intensity). Only three studies in the present review investigated the impact of different patterning of PA accrual; Andersson et al (2015) [31] and Parsons et al (2016) [49] investigated the impact of PA accrual in bouts of 10 minutes compared to non-bouts, and Vandercappellen et al., (2020) [55] investigated the influence of PA accrual on weekday vs weekend day [55]. Experimental studies have demonstrated that the acute effects of PA and exercise on arterial stiffness may vary depending on the pattern, intensity and duration of PA [15, 67, 68] Collectively this evidence suggests that the influence of a given volume of PA on arterial stiffness may not be uniform, rather it may depend on the between- and within-day variation in which PA is accumulated. Therefore, future research should adopt an event-based approach to analysis rather than rely on a single aggregate measure of PA [69].

## Strengths and limitations

The present review aimed to build on and overcome the limitations in previous work in this area through 1) the exclusion of studies investigating specific disease states, and 2) stipulating the inclusion of "standard" covariates in analytical models for inclusion in pooled analysis, both of which have been shown to affect associations with arterial stiffness [20–22] and thus limit the applicability of previous work. Additionally, the inclusion of studies involving both self-report and device-based measurements of PA is a strength of this review given that it allowed inclusion of a broader range of relevant literature than previous reviews and also highlighted how associations vary with measurement method. A larger number of studies also permitted subgroup analysis. However, this review is not without limitation. The heterogeneity in assessment of PA and subsequently the wide range of PA exposure metrics reported within

the literature limited the extent to which meta-analyses could be conducted, limiting the ability to draw firm conclusions. In order to provide a quantitative assessment of all included studies a vote count approach was employed; this approach is limited in that each reported analysis is given the same weighting based on the direction of observed associations between PA and cfPWV regardless of the strength of association, sample size or quality. In contrast studies reporting multiple analyses contribute more weight towards vote-count totals, irrespective of whether these analyses are independent of one another. Nevertheless, the need for this approach to summarise the available literature is an important reflection in itself, and highlights the methodological heterogeneity within current literature and reinforces the need for high quality research, standardised procedures and best-practice guidance for assessing and reporting PA in studies concerned with vascular function and CVD risk [66, 70].

## Conclusion

This meta-analysis and narrative synthesis provides evidence that more hPA is beneficially associated with lower (better) cfPWV, which is now considered an important indicator of CVD risk. These findings are broadly consistent with existing evidence for the beneficial effect of hPA on well-established risk markers for development of cardiovascular disease. The heterogeneity in PA measurement methods and study quality means that findings should be interpreted with a degree of caution. The development of methods to precisely quantify day-to-day movement behaviours should support future high-quality research in this field.

## Supporting information

**S1 File. Base search strategy on MEDLINE database.**
(PDF)

**S1 Fig. Forest plot of pooled analysis indicating the association between habitual moderate-vigorous or total physical activity and carotid-femoral pulse wave velocity.** Random effects model. Grey squares indicate individual study association with 95% confidence intervals (CI). Solid vertical line represents line of no difference. Dashed vertical line and solid black diamond represents the overall summary estimate of association.
(TIF)

**S2 Fig. Leave-one-out sensitivity analyses displaying the resultant overall association when one study at a time is sequentially removed from meta-analysis.** Grey dashed line indicates the overall effect size with all included studies (N = 18) as per main forest plot. Black circles indicate the resultant effect size (N = 17) when the current study is removed from the meta-analysis, with 95% confidence intervals. The overall association was not significantly changed by the removal of any one study.
(TIF)

**S3 Fig. Exploratory forest plot investigating the removal of the most influential study identified in the leave-one-out analysis.** Random effects model. Grey squares indicate individual study association with 95% confidence intervals (CI). Solid vertical line represents line of no difference. Dashed vertical line and solid black diamond represents the overall summary estimate of association.
(TIF)

**S4 Fig. Funnel plot and Egger test to investigate risk of publication bias.** No small study effects were found.
(TIF)

**S5 Fig. Forest plot of pooled analysis without Authors' manual conversion of univariate r values to partial r, indicating the association between habitual moderate-vigorous or total physical activity and carotid-femoral pulse wave velocity, grouped by those reporting univariate r and partial r.** Top indicates univariate r as reported in publication and bottom indicates partial r adjusted for any number of covariates as reported in the publication. Random effects model. Grey squares indicate individual study association with 95% confidence intervals (CI). Solid vertical line represents line of no difference. Dashed vertical line and grey diamond represents the overall summary estimate of association.
(TIF)

**S6 Fig. Forest plot of pooled analysis without Authors' manual conversion of univariate r values to partial r, grouped by those utilising accelerometer and those utilising questionnaire in the measurement of habitual physical activity.** Random effects model. Grey squares indicate individual study association with 95% confidence intervals (CI). Solid vertical line represents line of no difference. Dashed vertical line and black diamond represents the overall summary estimate of association.
(TIF)

**S1 Table. Full data extraction summary for all 29 included studies.**
(PDF)

## Acknowledgments

R.Lear would like to thank the Funds For Women Graduates for their support throughout her PhD.

## Author Contributions

**Conceptualization:** Rebecca Lear, Richard Pulsford.

**Data curation:** Rebecca Lear, Gemma Brailey.

**Formal analysis:** Rebecca Lear, Brad Metcalf.

**Investigation:** Rebecca Lear.

**Methodology:** Rebecca Lear, Brad Metcalf, Michael Nunns, Richard Pulsford.

**Project administration:** Rebecca Lear.

**Supervision:** Melvyn Hillsdon, Richard Pulsford.

**Validation:** Rebecca Lear.

**Visualization:** Rebecca Lear.

**Writing – original draft:** Rebecca Lear.

**Writing – review & editing:** Rebecca Lear, Brad Metcalf, Michael Nunns, Bert Bond, Melvyn Hillsdon, Richard Pulsford.

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
