## [Decision Letter · Decision Letter 0]

14 Sep 2022

PONE-D-22-16539Associations of habitual physical activity and carotid-femoral pulse wave velocity; a systematic review and meta-analysis of observational studies.PLOS ONE

Dear Dr. Lear,

Thank you for submitting your manuscript to PLOS ONE. After careful consideration, we feel that it has merit but does not fully meet PLOS ONE’s publication criteria as it currently stands. Therefore, we invite you to submit a revised version of the manuscript that addresses all the points raised during the review process.Please submit your revised manuscript by Oct 29 2022 11:59PM. If you will need more time than this to complete your revisions, please reply to this message or contact the journal office at plosone@plos.org. Please include the following items when submitting your revised manuscript:A rebuttal letter that responds to each point raised by the academic editor and reviewer(s). You should upload this letter as a separate file labeled 'Response to Reviewers'.A marked-up copy of your manuscript that highlights changes made to the original version. You should upload this as a separate file labeled 'Revised Manuscript with Track Changes'.An unmarked version of your revised paper without tracked changes. You should upload this as a separate file labeled 'Manuscript'.

We look forward to receiving your revised manuscript.

Kind regards,

Laurent Mourot

Section Editor

PLOS ONE

Journal Requirements:

Reviewers' comments:

Reviewer's Responses to Questions

**Comments to the Author**

1. Is the manuscript technically sound, and do the data support the conclusions?

Reviewer #1: Yes

2. Has the statistical analysis been performed appropriately and rigorously? 

Reviewer #1: I Don't Know

3. Have the authors made all data underlying the findings in their manuscript fully available?

Reviewer #1: Yes

4. Is the manuscript presented in an intelligible fashion and written in standard English?

Reviewer #1: Yes

5. Review Comments to the Author

Reviewer #1: The introduction begins with information about the problem of cardiovascular disease (CVD), especially its impact on the public health system. Then the protective effects of exercise, recommendations, and the inverse relationship between CVD and exercise. The third paragraph showed information about other cardiovascular risk factors but that do not alter the inverse relationship between exercise and CVD. However, the sequence of ideas in the fourth and fifth paragraph is confusing. Information about vascular health appears, then about exercise and then the different methods of vascular outcome measurement. It would be interesting to organize the ideas in paragraphs 4 and 5 for a better understanding of the study's problem.

In line 165 “Authors were contacted to obtain this information if not reported in the paper (N=14)”, Does the number refer to the authors contacted or to the number of information not obtained?

In line 176. “In eight studies30 33-40 univariate” the number of the references does not correspond to the information described.

In line 206. “Five studies34 44-46 included a group” revise the number of references.

In line 214. 16 measured hPA via self-report questionnaire or interview.27 30 34-36 38-40 42 43 46 48 50 56-58

In figure 1, if we consider all the excluded reports from the reports assessed for eligibility, 13 studies remain. Probably one study fell into more than one exclusion criterion. It would be clearer to describe this information in the results.

6. PLOS authors have the option to publish the peer review history of their article (what does this mean?). If published, this will include your full peer review and any attached files.

Reviewer #1: No

---

## [Author Response · Author response to Decision Letter 0]

28 Sep 2022

Author response: Thank you. We have been through the PLoS ONE style requirements and have made edits to ensure it meets the required standard. 

Author Response: No data has been removed, rather 3 new supplementary files have now been added to the manuscript.

Author Response: We have now added a “supporting information” section at end of manuscript with associated captions. 

Comments to the Author

5. Review Comments to the Author

Reviewer #1: The introduction begins with information about the problem of cardiovascular disease (CVD), especially its impact on the public health system. Then the protective effects of exercise, recommendations, and the inverse relationship between CVD and exercise. The third paragraph showed information about other cardiovascular risk factors but that do not alter the inverse relationship between exercise and CVD. However, the sequence of ideas in the fourth and fifth paragraph is confusing. Information about vascular health appears, then about exercise and then the different methods of vascular outcome measurement. It would be interesting to organize the ideas in paragraphs 4 and 5 for a better understanding of the study's problem. 

Author response: Thank you for this comment, we welcome the opportunity to clarify this section. To improve the flow of the latter part of the introduction and clarify the rationale for the review we have made a number of edits and have moved the discussion of vascular function and its assessment to earlier in the piece. This is now followed by discussion of evidence for associations between structured exercise and vascular function, and the evidence gaps which this review addresses, namely the lack of clarity regarding associations between habitual physical activity and arterial stiffness and the limitations of existing reviews.

In line 165 “Authors were contacted to obtain this information if not reported in the paper (N=14)”, Does the number refer to the authors contacted or to the number of information not obtained? 

Author response: Thank you, we appreciate the opportunity to clarify. The text has now been edited to read: “We contacted the corresponding authors of 14 studies to request any information required for meta-analysis that was not reported in their papers." 

The resultant information not obtained from contacting Authors and therefore necessitating manual conversion is then detailed in the following bullet points 1-3, together with citations of the individual studies. 

In line 176. “In eight studies30 33-40 univariate” the number of the references does not correspond to the information described. 

Author response: Thank-you for pointing out this error. These references have been checked and the extra reference (30) removed. 

In line 206. “Five studies34 44-46 included a group” revise the number of references. 

Author response: Thank-you for pointing out this error. There were only four studies included a subset of participants with disease. These are specified correctly. The text has been edited to describe the “four” rather than ‘five’ studies. 

In line 214. 16 measured hPA via self-report questionnaire or interview.27 30 34-36 38-40 42 43 46 48 50 56-58. 

Author Response: These references have been double checked and are indeed correct. No changes have been made. 

In figure 1, if we consider all the excluded reports from the reports assessed for eligibility, 13 studies remain. Probably one study fell into more than one exclusion criterion. It would be clearer to describe this information in the results. 

Author response: This is correct, we appreciate the opportunity to clarify. Several studies were excluded based on them meeting more than one exclusion criteria. This has been addressed by the addition of “(not mutually exclusive)” within figure 1 for clarity. 

Additionally, this information has now been added in full in lines 206-212 (clean manuscript file, or lines 218-225 in track changes manuscript file) as follows: “Database searches yielded 8149 studies of which 180 full-texts were screened for eligibility. 151 studies were excluded for the following reasons: Investigating clinical population without healthy control group (N=21), Measuring exercise only with no measurement of habitual PA (N=50), alternative measurement of vascular function without cfPWV (N=51), insufficient data reported (N=37) and the inadequate categorization of participants restricting analysis (N=8). Note, a number of studies were excluded for meeting more than one of these exclusion criteria. A PRISMA flow diagram is presented in Fig 1.”

---

## [Decision Letter · Decision Letter 1]

17 Oct 2022

PONE-D-22-16539R1Associations of habitual physical activity and carotid-femoral pulse wave velocity; a systematic review and meta-analysis of observational studies.PLOS ONE

Dear Dr. Lear,

Thank you for submitting your manuscript to PLOS ONE. After careful consideration, we feel that it has merit but does not fully meet PLOS ONE’s publication criteria as it currently stands. Therefore, we invite you to submit a revised version of the manuscript that addresses the points raised during the review process. Important limitations still persist, especially specifying how physical activity (PA) was measured and if it has an importance in the link between PA and pulse wave velocity, and what is the effect of duration/intensity/total work on this link / these links. The difference/concordance with the litterature is also of importance. This is at the heart of the manuscript and should be taken into account in your answers. Please submit your revised manuscript by Dec 01 2022 11:59PM. If you will need more time than this to complete your revisions, please reply to this message or contact the journal office at plosone@plos.org. Please include the following items when submitting your revised manuscript:A rebuttal letter that responds to each point raised by the academic editor and reviewer(s). You should upload this letter as a separate file labeled 'Response to Reviewers'.A marked-up copy of your manuscript that highlights changes made to the original version. You should upload this as a separate file labeled 'Revised Manuscript with Track Changes'.An unmarked version of your revised paper without tracked changes. You should upload this as a separate file labeled 'Manuscript'.

We look forward to receiving your revised manuscript.

Kind regards,

Laurent Mourot

Section Editor

PLOS ONE

Reviewers' comments:

Reviewer's Responses to Questions

**Comments to the Author**

1. If the authors have adequately addressed your comments raised in a previous round of review and you feel that this manuscript is now acceptable for publication, you may indicate that here to bypass the “Comments to the Author” section, enter your conflict of interest statement in the “Confidential to Editor” section, and submit your "Accept" recommendation.

Reviewer #1: All comments have been addressed

2. Is the manuscript technically sound, and do the data support the conclusions?

Reviewer #1: Yes

3. Has the statistical analysis been performed appropriately and rigorously? 

Reviewer #1: Yes

4. Have the authors made all data underlying the findings in their manuscript fully available?

Reviewer #1: Yes

5. Is the manuscript presented in an intelligible fashion and written in standard English?

Reviewer #1: Yes

6. Review Comments to the Author

Reviewer #1: Thank you for taking my considerations into account. I still have some questions for the authors.

The purpose of this present review is to clarify the associations between habitual physical activity and arterial stiffness, responding to the limits of a previous review. This information appears in the introduction and there is nothing further. It would be interesting to have an approach on both studies, the previous review and the current one in the discussion.

If I understand correctly, data adjusted for age, sex, BMI and BP were used. However, the units of measurement of physical activity are extremely variable. In the authors' opinion, is it possible that the association between habitual physical activity and PWV was not evident from these differences? Perhaps trying to classify the studies according to the amount of physical activity and/or intensity would reveal a stronger association. Explore more the information of the lines 388-393.

In the discussion between the lines the authors cite physical training. The comparison cited between the results of this review and previous studies of physical training needs to be approached with caution. First, because of the limitations cited above regarding self-reporting of physical activity. Second, because this is not the initial purpose of this review. Finally, because the results of this review are not the effects of interventional studies of physical training. However, exploring why the results are consistent across studies despite differences in modalities may be interesting and may help answer my previous question.

7. PLOS authors have the option to publish the peer review history of their article (what does this mean?). If published, this will include your full peer review and any attached files.

Reviewer #1: No

---

## [Author Response · Author response to Decision Letter 1]

30 Nov 2022

We thank the editor and reviewer for taking the time to read our manuscript and for their useful and constructive comments. Please find our responses below.

1) The purpose of this present review is to clarify the associations between habitual physical activity and arterial stiffness, responding to the limits of a previous review. This information appears in the introduction and there is nothing further. It would be interesting to have an approach on both studies, the previous review and the current one in the discussion.

Author Response: Thank you for this comment, we agree that it is useful to compare the results from the present review with that of the previous review, on which we are providing an update. The text in the first paragraph of the discussion has now been amended to compare the results from our own pooled analyses with that of the previous review. 

2) If I understand correctly, data adjusted for age, sex, BMI and BP were used. However, the units of measurement of physical activity are extremely variable. In the authors' opinion, is it possible that the association between habitual physical activity and PWV was not evident from these differences? Perhaps trying to classify the studies according to the amount of physical activity and/or intensity would reveal a stronger association. Explore more the information of the lines 388-393.

Author Response: Thank you for your comment. We agree that the measurement units of habitual physical activity (hPA) were very variable within included studies. It is for this reason that we converted all association results to partial r or standardised beta – by doing this we are instead reporting the association between per standard deviation change in hPA, rather than in absolute units (of which are variable). This allowed us to make direct comparisons across different studies that have measured and reported hPA in different ways. 

The amount of physical activity and the intensity of physical activity was considered when completing our analyses. A subgroup analysis was completed on studies reporting associations with total PA and compared to those reporting associations with MVPA (figure 3 in the manuscript). However unfortunately it was not possible to complete separate pooled analysis for associations with light intensity activity due to the limited number of studies reporting this association (N=6) and the lack of data available from these six studies meaning it was not possible to convert the available association statistics to partial r / standardised beta for comparison across studies. 

Similarly, we decided it would not be appropriate to complete a subgroup analysis, or meta-regression on the average level of PA completed by the cohort within each study due to the limited number of included studies that reported average PA level (N=9 out of 18) and the differing units of PA measurement reported within these 9 studies. 

3) In the discussion between the lines the authors cite physical training. The comparison cited between the results of this review and previous studies of physical training needs to be approached with caution. First, because of the limitations cited above regarding self-reporting of physical activity. Second, because this is not the initial purpose of this review. Finally, because the results of this review are not the effects of interventional studies of physical training. However, exploring why the results are consistent across studies despite differences in modalities may be interesting and may help answer my previous question. 

Author Response: We thank you for your comment and agree it is not useful to compare the results from the present review to those of previous meta-analyses on exercise training interventions. This text has now been edited to remove reference of training intervention studies and instead provide a detailed comparison with that of the previous systematic review on habitual PA and cfPWV.

---

## [Decision Letter · Decision Letter 2]

27 Mar 2023

Associations of habitual physical activity and carotid-femoral pulse wave velocity; a systematic review and meta-analysis of observational studies.

PONE-D-22-16539R2

Dear Dr. Lear,

We’re pleased to inform you that your manuscript has been judged scientifically suitable for publication and will be formally accepted for publication once it meets all outstanding technical requirements.

Kind regards,

Laurent Mourot

Section Editor

PLOS ONE

Additional Editor Comments (optional):

Reviewers' comments:

Reviewer's Responses to Questions

**Comments to the Author**

1. If the authors have adequately addressed your comments raised in a previous round of review and you feel that this manuscript is now acceptable for publication, you may indicate that here to bypass the “Comments to the Author” section, enter your conflict of interest statement in the “Confidential to Editor” section, and submit your "Accept" recommendation.

Reviewer #1: All comments have been addressed

2. Is the manuscript technically sound, and do the data support the conclusions?

Reviewer #1: Yes

3. Has the statistical analysis been performed appropriately and rigorously? 

Reviewer #1: Yes

4. Have the authors made all data underlying the findings in their manuscript fully available?

Reviewer #1: Yes

5. Is the manuscript presented in an intelligible fashion and written in standard English?

Reviewer #1: Yes

6. Review Comments to the Author

Reviewer #1: The study is relevant and the method followed the PRISMA recommendations. The statistical analysis is robust and takes into account factors that influence the main measurement variable. The suggestions made were accepted and the necessary changes were made.

7. PLOS authors have the option to publish the peer review history of their article (what does this mean?). If published, this will include your full peer review and any attached files.

Reviewer #1: **Yes: **Daniele Peres

---

## [Editor Report · Acceptance letter]

30 Mar 2023

PONE-D-22-16539R2 

Associations of habitual physical activity and carotid-femoral pulse wave velocity; a systematic review and meta-analysis of observational studies. 

Dear Dr. Lear:

I'm pleased to inform you that your manuscript has been deemed suitable for publication in PLOS ONE. Congratulations! Your manuscript is now with our production department. 

Kind regards, 

on behalf of

Dr Laurent Mourot 

Section Editor

PLOS ONE